# TRANS-ENCODER:
# UNSUPERVISED SENTENCE-PAIR MODELLING
# THROUGH SELF- AND MUTUAL-DISTILLATIONS

**Fangyu Liu**[1]* **Yunlong Jiao**[2] **Jordan Massiah**[2] **Emine Yilmaz**[2] **Serhii Havrylov**[2]
[1]University of Cambridge  [2]Amazon
`fl399@cam.ac.uk` `{jyunlong,jormas,eminey,havrys}@amazon.com`

## ABSTRACT

In NLP, a large volume of tasks involve pairwise comparison between two sequences (e.g., sentence similarity and paraphrase identification). Predominantly, two formulations are used for sentence-pair tasks: bi-encoders and cross-encoders. Bi-encoders produce fixed-dimensional sentence representations and are computationally efficient, however, they usually underperform cross-encoders. Cross-encoders can leverage their attention heads to exploit inter-sentence interactions for better performance but they require task finetuning and are computationally more expensive. In this paper, we present a completely unsupervised sentence-pair model termed as TRANS-ENCODER that combines the two learning paradigms into an iterative joint framework to simultaneously learn enhanced bi- and cross-encoders. Specifically, on top of a pre-trained language model (PLM), we start with converting it to an unsupervised bi-encoder, and then alternate between the bi- and cross-encoder task formulations. In each alternation, one task formulation will produce pseudo-labels which are used as learning signals for the other task formulation. We then propose an extension to conduct such self-distillation approach on multiple PLMs in parallel and use the average of their pseudo-labels for mutual-distillation. TRANS-ENCODER creates, to the best of our knowledge, the first completely unsupervised cross-encoder and also a state-of-the-art unsupervised bi-encoder for sentence similarity. Both the bi-encoder and cross-encoder formulations of TRANS-ENCODER outperform recently proposed state-of-the-art unsupervised sentence encoders such as Mirror-BERT (Liu et al., 2021) and SimCSE (Gao et al., 2021) by up to $5\%$ on the sentence similarity benchmarks. Code and models are released at `https://github.com/amzn/trans-encoder`.

## 1 INTRODUCTION

Comparing pairwise sentences is fundamental to a wide spectrum of tasks in NLP such as information retrieval (IR), natural language inference (NLI), semantic textual similarity (STS) and clustering. Two general architectures usually used for sentence-pair modelling are bi-encoders and cross-encoders.

In a cross-encoder, two sequences are concatenated and sent into the model (usually deep Transformers like BERT/RoBERTa; Devlin et al. 2019; Liu et al. 2019) in one pass. The attention heads of Transformers could directly model the inter-sentence interactions and output a classification/relevance score. However, a cross-encoder needs to recompute the encoding for different combinations of sentences in each unique sequence pair, resulting in a heavy computational overhead. It is thus impractical in tasks like IR and clustering where massive pairwise sentence comparisons are involved. Also, task finetuning is always required for converting PLMs to cross-encoders. By contrast, in a bi-encoder, each sequence is encoded separately and mapped to a common embedding space for similarity comparison. The encoded sentences can be cached and reused. A bi-encoder is thus much more efficient. Also, the output of a bi-encoder can be used off-the-shelf as sentence embeddings for other downstream tasks. That said, it is well-known that in supervised learning, bi-encoders underperform cross-encoders (Humeau et al., 2020; Thakur et al., 2021) since the former could not

---

*Work done during internship at Amazon.

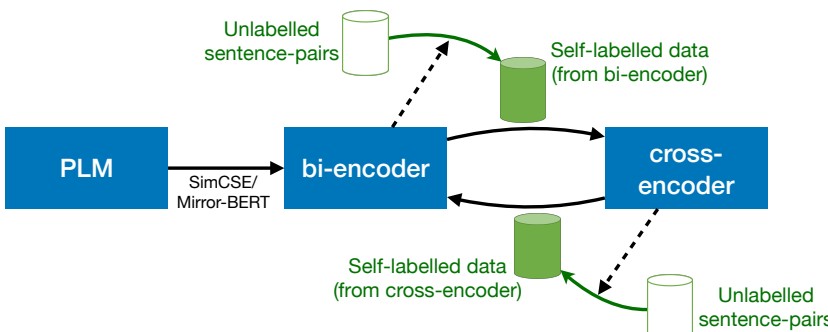

Figure 1: A graphical illustration of the self-distillation learning scheme in TRANS-ENCODER. Notice that the blue boxes represent the same model architecture trained sequentially.

explicitly model the interactions between sentences but could only compare them in the embedding space in a *post hoc* manner.

In this work, we ask the question: can we leverage the advantages of both bi- and cross-encoders and bootstrap knowledge from them in an unsupervised manner? Our proposed TRANS-ENCODER addresses this question with the following intuition: As a starting point, we can use bi-encoder representations to tune a cross-encoder. With more powerful inter-sentence modelling, the cross-encoder should resurface more knowledge from the PLMs than the bi-encoder given the same data. In turn, the more powerful cross-encoder can distil its knowledge back to the bi-encoder. We can repeat this cycle to iteratively bootstrap from both the bi- and cross-encoders.

## 2  TRANS-ENCODER

The general idea of TRANS-ENCODER is simple yet extremely effective. In §2.1, we first transform an off-the-shelf PLM to a strong bi-encoder, serving as an initialisation point. Then, the bi-encoder produces pseudo-labels and the PLM subsequently learns from these pseudo-labels in a cross-encoder manner (§2.2). Consecutively, the cross-encoder further produces more accurate pseudo-labels for bi-encoder learning (§2.3). This self-distillation process is visualised in Fig. 1. Then in, §2.4, we propose a further extension called mutual-distillation that stabilises the training process and boosts the encoder performance even more.

### 2.1  TRANSFORM PLMS INTO EFFECTIVE BI-ENCODERS

Off-the-shelf PLMs are unsatisfactory bi-encoders.[1] To have a reasonably good starting point, we leverage a simple contrastive tuning procedure to transform existing PLMs to bi-encoders. This approach is concurrently proposed in both (Liu et al., 2021) and (Gao et al., 2021).

Let $f(\cdot)$ denote the encoder model (a Transformer-based model); $\mathcal{X}$ be a batch of randomly sampled raw sentences. For any sentence $x_i \in \mathcal{X}$, we send it to the encoder twice to create two views of the same data point: $f(x_i)$ and $f(\overline{x}_i)$. Note that the two encodings slightly differ due to the dropout layers in the encoder.[2] We then use the standard InfoNCE loss (Oord et al., 2018) to cluster the positive pairs together and push away the negative pairs in the mini-batch:

$$\mathcal{L}_{\text{infoNCE}} = -\sum_{i=1}^{|\mathcal{X}|} \log \frac{\exp(\cos(f(x_i), f(\overline{x}_i))/\tau)}{\sum_{x_j \in \mathcal{N}_i} \exp(\cos(f(x_i), f(x_j))/\tau)}. \tag{1}$$

$\tau$ denotes a temperature parameter; $\mathcal{N}_i$ denotes all positive & negatives of $x_i$ within the current data batch. Intuitively, the numerator is the similarity of the self-duplicated pair (the positive example)

---

[1]In this work, a *bi-encoder* is the same as a *universal text encoder*. When embedding a pair of sentences under the bi-encoder architecture, the same encoder is used (i.e., the two-branches of bi-encoder share weights).

[2]Mirror-BERT has the option to also randomly mask a span in $\overline{x}_i$ and using drophead (Zhou et al., 2020) to replace dropout as hidden states augmentation.

and the denominator is the sum of the similarities between $x_i$ and all other negative strings besides $\overline{x}_i$. The loss encourages the positive pairs to be relatively close compared to the negative ones. In experiments, we use checkpoints released by SimCSE and Mirror-BERT. But in principle, any techniques could be used here as long as the method serves the purpose of transforming BERT to an effective bi-encoder and does not require additional labelled data (see §6).

## 2.2 SELF-DISTILLATION: BI- TO CROSS-ENCODER

After obtaining a sufficiently good bi-encoder, we leverage it to label sentence pairs sampled from the task of interest. Specifically, for a given sentence-pair (sent1, sent2), we input them to the bi-encoder separately and get two embeddings (we use the embedding of [CLS] from the last layer). The cosine similarity between them is regarded as their relevance score. In this way we have constructed a self-labelled sentence-pair scoring dataset in the format of (sent1, sent2, score).

We then employ the same model architecture to learn from these score, but with a cross-encoder formulation. The cross-encoder weights are initialised from the original PLM.[3] For the sentence-pair (sent1, sent2), we concatenate them to produce "[CLS] sent1 [SEP] sent2 [SEP]" and input it to the cross-encoder. A linear layer (newly initialised) then map the sequence's encoding (embedding of the [CLS] token) to a scalar. The learning objective of the cross-encoder is minimising the KL divergence between its predictions and the self-labelled scores from the bi-encoder. This is equivalent to optimising the (soft) binary cross-entropy (BCE) loss:

$$\mathcal{L}_{\text{BCE}} = -\frac{1}{N} \sum_{n=1}^{N} \Big( y_n \cdot \log(\sigma(x_n)) + (1 - y_n) \cdot \log(1 - \sigma(x_n)) \Big) \tag{2}$$

where $N$ is the data-batch size; $\sigma(\cdot)$ is the sigmoid activation; $x_n$ is the prediction of the cross-encoder; $y_n$ is the self-labelled ground-truth score from the bi-encoder.

Note that while the cross-encoder is essentially learning from the data produced by itself (in a bi-encoder form), usually, it outperforms the original bi-encoder on held-out data. The cross-encoder directly discovers the similarity between two sentences through its attention heads, finding more accurate cues to justify the relevance score. The ability of discovering such cues could then generalise to unseen data, resulting in stronger sentence-pair scoring capability than the original bi-encoder. From a knowledge distillation perspective, we can view the bi- and cross-encoder as the teacher and student respectively. In this case the student outperforms the teacher, not because of stronger model capacity, but smarter task formulation. By leveraging this simple yet powerful observation, we are able to design a learning scheme that iteratively boosts the performance of both bi- and cross-encoder.

## 2.3 SELF-DISTILLATION: CROSS- TO BI-ENCODER

With the more powerful cross-encoder at hand, a natural next step is distilling the extra gained knowledge back to a bi-encoder form, which is more useful for downstream tasks. Besides, and more importantly, a better bi-encoder could produce even more self-labelled data for cross-encoder learning. In this way we could repeat §2.2 and §2.3, continually bootstrapping the encoder performance.

We create the self-labelled sentence-scoring dataset in the same way as §2.2 except that the cross-encoder is used for producing the relevance score. The bi-encoder is initialised with the weights after SimCSE training from §2.1.[4] For every sentence pair, two sentence embeddings are produced separately by the bi-encoder. The cosine similarity between the two embeddings are regarded as their predictions of the relevance score. The aim is to regress the predictions to the self-labelled scores by the cross-encoder. We use a mean square error (MSE) loss:

$$\mathcal{L}_{\text{MSE}} = -\frac{1}{N} \sum_{n=1}^{N} \Big( x_n - y_n \Big)^2 \tag{3}$$

---

[3]We have a design choice to make here, we could either (1) directly use the weights of the bi-encoder or (2) use the weights of the original PLM. We will show in experiments (§5.2) that (2) is slightly better than (1).

[4]Again, we have the choice of using the cross-encoder weights from §2.2 (the extra linear layer is disregarded). We will discuss more in experiments.

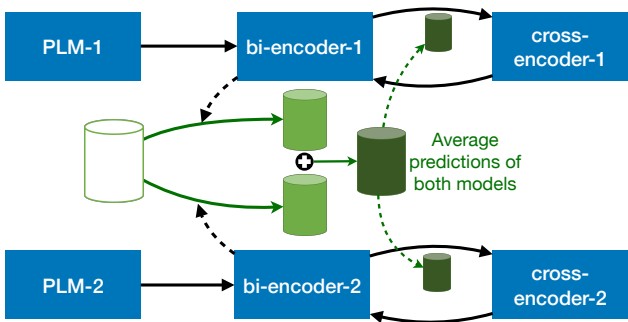

Figure 2: A graphical illustration of the mutual-distillation learning scheme in TRANS-ENCODER. Note that, for simplicity, only the bi- to cross-encoder mutual-distillation is shown. We also conduct cross- to bi-encoder mutual-distillation in the same manner.

where $N$ is the batch size; $x_n$ is the cosine similarity between a sentence pair; $y_n$ is the self-labelled ground-truth. In experiments, we will show that this resulting bi-encoder is more powerful than the initial bi-encoder. Sometimes, the bi-encoder will even outperform its teacher (i.e., the cross-encoder).

**Choice of loss functions: maintaining student-teacher discrepancy is the key.** Intuitively, MSE allows for more accurate distillations since it punishes any numerical discrepancies between predicted scores and labels on the instance level. However, in practice, we found that using MSE for bi-to-cross distillation aggravates the overfitting issue for cross-encoder: the cross-encoder, with its strong sentence-pair modelling capability, completely overfits to the mapping between concatenated sentence pairs and the pseudo scores. This elimination of discrepancy between model predictions and pseudo labels harms generalisation and the iterative learning cycle cannot continue (as the predicted scores by the student model will be the same as the teacher model). We thus use BCE loss for bi-to-cross since BCE is a more *forgiving* loss function compared with MSE. According to our gradient derivation (App. §A.2), BCE essentially is a temperature-sharpened version of MSE, which is more tolerant towards numerical discrepancies. This prevents cross-encoders from overfitting to the pseudo labels completely. Similar issue does not exist for the cross-to-bi distillation as for bi-encoders, two input sequences are separately encoded and the model does not easily overfit to the labels. We have a more thorough discussion in App. §A.2 and Tab. 8 to highlight the rationales of the current configuration.

### 2.4 MUTUAL-DISTILLATION

The aforementioned self-learning approach has a drawback: since the model regresses to its previous predictions, it tends to reinforce its errors. To mitigate this problem, we design a simple mutual-distillation approach to smooth out the errors/biases originated from PLMs. Specifically, we conduct self-distillation on multiple PLMs in parallel (for brevity, we use two in this paper: BERT and RoBERTa, however the framework itself is compatible with any number of PLMs). Each PLM does not communicate/synchronise with each other except when producing the self-labelled scores. In mutual-distillation, we use the average predictions of all models as the ground-truth for the next round of learning. A graphical illustration is shown in Fig. 2.[5]

In the following, we call all self-distillation models TRANS-ENCODER (or TENC for short); all mutual-distillation models TRANS-ENCODER-mutual (or TENC-mutual for short). Note that one TENC (-mutual) training produces one best bi-encoder, called TENC (-mutual) (bi), and one best cross-encoder, called TENC (-mutual) (cross), based on dev set. We report numbers of both.

## 3 EVALUATION TASKS

**Evaluation task: semantic textual similarity (STS).** Following prior works (Reimers & Gurevych, 2019; Liu et al., 2021; Gao et al., 2021), we consider seven STS datasets: SemEval STS 2012-2016

---

[5]We also experimented with a sequential ensemble strategy: initialising the cross- and bi-encoder with BERT/RoBERTa alternatively (1 cycle of BERT then 1 cycle of RoBERTa) but found the performance worse than mutual-distillation.

(STS12-16, Agirre et al. 2012; 2013; 2014; 2015; 2016), STS Benchmark (STS-B, Cer et al. 2017) and SICK-Relatedness (SICK-R, Marelli et al. 2014). In all these datasets, sentence pairs are given a human-judged relevance score from 1 to 5. We normalise them to 0 and 1 and models are asked to predict the scores. We report Spearman's $\rho$ rank correlation between the two.

**Evaluation task: binary classification.** We also test TRANS-ENCODER on sentence-pair binary classification tasks where the model has to decide if a sentence-pair has certain relations. We choose (1) the Quora Question Pair (QQP) dataset, requiring a model to judge whether two questions are duplicated; (2) a question-answering entailment dataset QNLI (Rajpurkar et al., 2016; Wang et al., 2019) in which given a question and a sentence the model needs to judge if the sentence answers the question; (3) the Microsoft Research Paraphrase Corpus (MRPC) which asks a model to decide if two sentences are paraphrases of each other. The ground truth labels of all datasets are either 0 or 1. Following (Li et al., 2020; Liu et al., 2021), we compute Area Under Curve (AUC) scores using the binary labels and the relevance scores predicted by models.

## 4 EXPERIMENTAL SETUP

**Training and evaluation details.** For each task, we use all available sentence pairs (from train, development and test sets of all datasets combined) without their labels as training data. The original QQP and QNLI datasets are extremely large. We thus downsample QQP to have 10k, 1k and 10k pairs for train, dev and test; QNLI to have 10k train set. QNLI does not have public ground truth labels for testing. So, we use its first 1k examples in the official dev set as our dev data and the rest in the official dev set as test data. The dev set for MRPC is its official dev sets. The dev set for STS12-16, STS-B and SICK-R is the dev set of STS-B. We save one checkpoint for every 200 training steps and at the end of each epoch. We use the dev sets to select the best model for testing.[6] Dev sets are also used to tune the hyperprameters in each task.

For clear comparison with SimCSE and Mirror-BERT, we use their released checkpoints as initialisation points (i.e., we do not train models in §2.1 ourselves). We consider four SimCSE variants. Two base variants: SimCSE-BERT-base, SimCSE-RoBERTa-base; and two large variants: SimCSE-BERT-large, SimCSE-RoBERTa-large. We consider two Mirror-BERT variants: Mirror-RoBERTa-base and Mirror-RoBERTa-base-drophead.[7] For brevity, our analysis in the main text focuses on SimCSE models. We list Mirror-BERT results in Appendix. We train TRANS-ENCODER models for 3 cycles on the STS task and 5 cycles on the binary classification tasks. Within each cycle, all bi- and cross-encoders are trained for 10 and 1 epochs respectively for the STS task; 15 and 3 epochs for binary classification.[8] All models use AdamW (Loshchilov & Hutter, 2019) as the optimiser. In all tasks, unless noted otherwise, we create final representations using `[CLS]`. We train our base models on a server with 4 * V100 (16GB) GPUs and large models on a server with 8 * A100 (40GB) GPUs. All main experiments have the same fixed random seed. All other hparams are listed in Appendix.

**Mutual-distillation setup.** For the two models used for mutual-distillation, they are either (1) the base variant of SimCSE-BERT and SimCSE-RoBERTa or (2) the large variant of the two. Theoretically, we could mutually distil even more models but we keep the setup simple for fast and clear comparison. Also, since mutual-distillation models use information from both PLMs, we also list (in tables) *ensemble results* that use the average of the predictions from two TRANS-ENCODERs.

## 5 RESULTS AND DISCUSSION

### 5.1 MAIN RESULTS

**STS results (Tab. 1).** The main results for STS are listed in Tab. 1. Compared with the baseline SimCSE, TRANS-ENCODER has brought significant improvements across the board. With various variants of the SimCSE as the base model, TRANS-ENCODER consistently enhances the average score

---

[6]Note that checkpoints of bi- and cross-encoders are independently saved and evaluated on dev set. In the end, one best bi-encoder and one best cross-encoder are obtained.

[7]We do not consider BERT-based checkpoints by Liu et al. (2021) since they adopt mean pooling over all tokens. For brevity, we only experiment with encoders trained with `[CLS]`.

[8]We use fewer epochs for cross-encoders since they usually converge much faster than bi-encoders.

| # | dataset→ | STS12 | STS13 | STS14 | STS15 | STS16 | STS-B | SICK-R | avg. |
|---|---|---|---|---|---|---|---|---|---|
| | *single-model results* | | | | | | | | |
| 1 | SimCSE-BERT-base* | 68.64 | 82.39 | 74.30 | 80.69 | 78.71 | 76.54 | 72.22 | 76.21 |
| 1.1 | + TENC (bi) | 72.17 | 84.40 | 76.69 | 83.28 | 80.91 | 81.26 | 71.84 | 78.65 |
| 1.2 | + TENC (cross) | 71.94 | 84.14 | 76.39 | 82.87 | 80.65 | 81.06 | 71.16 | 78.32 |
| 1.3 | + TENC-mutual (bi) | 75.09 | 85.10 | 77.90 | **85.08** | 83.05 | 83.90 | **72.76** | **80.41** |
| 1.4 | + TENC-mutual (cross) | **75.44** | **85.59** | **78.03** | 84.44 | 82.65 | 83.61 | 69.52 | 79.90 |
| 2 | SimCSE-RoBERTa-base | 68.34 | 80.96 | 73.13 | 80.86 | 80.61 | 80.20 | 68.62 | 76.10 |
| 2.1 | + TENC (bi) | 73.36 | 82.47 | 76.39 | 83.96 | 82.67 | 82.05 | 67.63 | 78.36 |
| 2.2 | + TENC (cross) | 72.59 | 83.24 | 76.83 | 84.20 | 82.82 | 82.85 | 69.51 | 78.86 |
| 2.3 | + TENC-mutual (bi) | 75.01 | 85.22 | 78.26 | 85.16 | 83.22 | 83.88 | **72.56** | 80.47 |
| 2.4 | + TENC-mutual (cross) | **76.37** | **85.87** | **79.03** | **85.77** | **83.77** | **84.65** | 72.62 | **81.15** |
| 3 | SimCSE-BERT-large | 71.30 | 84.32 | 76.32 | 84.28 | 79.78 | 79.04 | **73.88** | 78.42 |
| 3.1 | + TENC (bi) | 75.55 | 84.08 | 77.01 | 85.43 | 81.37 | 82.88 | 71.46 | 79.68 |
| 3.2 | + TENC (cross) | 75.81 | 84.51 | 76.50 | 85.65 | 82.14 | 83.47 | 70.90 | 79.85 |
| 3.3 | + TENC-mutual (bi) | **78.19** | **88.51** | **81.37** | **88.16** | **84.81** | **86.16** | 71.33 | **82.65** |
| 3.4 | + TENC-mutual (cross) | 77.97 | 88.31 | 81.02 | 88.11 | 84.40 | 85.95 | 71.92 | 82.52 |
| 4 | SimCSE-RoBERTa-large | 71.40 | 84.60 | 75.94 | 84.36 | 82.22 | 82.67 | 71.23 | 78.92 |
| 4.1 | + TENC (bi) | 77.92 | 86.69 | 79.29 | 87.23 | 84.22 | 86.10 | 68.36 | 81.40 |
| 4.2 | + TENC (cross) | **78.32** | 86.20 | 79.61 | 86.88 | 82.93 | 84.48 | 67.90 | 80.90 |
| 4.3 | + TENC-mutual (bi) | 78.15 | **88.39** | 81.76 | 88.38 | 84.95 | 86.55 | 72.31 | **82.93** |
| 4.4 | + TENC-mutual (cross) | 78.28 | 88.31 | **81.94** | **88.63** | **85.03** | **86.70** | 71.63 | **82.93** |
| | *ensemble results* (average predictions of two models) | | | | | | | | |
| 1+2 | SimCSE-base ensemble | 70.71 | 83.49 | 76.45 | 83.13 | 81.79 | 81.51 | 71.94 | 78.43 |
| 1.1+2.1 | TENC-base (bi) ensemble | 73.58 | 85.01 | 78.35 | 85.02 | 83.21 | 84.07 | 70.93 | 80.03 |
| 1.2+2.2 | TENC-base (cross) ensemble | 74.25 | 84.92 | 78.57 | 85.16 | 83.25 | 83.52 | 70.73 | 80.06 |
| 1.3+2.3 | TENC-mutual-base (bi) ensemble | 75.29 | 85.34 | 78.49 | 85.28 | 83.43 | 84.09 | **72.75** | 80.67 |
| 1.4+2.4 | TENC-mutual-base (cross) ensemble | **76.60** | **86.18** | **79.09** | **85.49** | **83.64** | **84.67** | 71.96 | **81.09** |
| 3+4 | SimCSE-large ensemble | 73.22 | 86.03 | 78.15 | 85.95 | 82.83 | 83.05 | **73.86** | 80.66 |
| 3.1+4.1 | TENC-large (bi) ensemble | 78.49 | 87.02 | 80.31 | 87.85 | 84.31 | **86.54** | 72.39 | 82.41 |
| 3.2+4.2 | TENC-large (cross) ensemble | 77.93 | 86.91 | 79.83 | 87.82 | 83.74 | 85.58 | 72.02 | 81.98 |
| 3.3+4.3 | TENC-mutual-large (bi) ensemble | 78.42 | 88.60 | **81.91** | 88.38 | **85.01** | 86.52 | 72.23 | 83.01 |
| 3.4+4.4 | TENC-mutual-large (cross) ensemble | **78.52** | **88.70** | 81.90 | **88.67** | 84.96 | 86.70 | 72.03 | **83.07** |

Table 1: English STS. Spearman's $\rho$ rank correlations are reported. TENC models use only self-distillation while TENC-mutual models use mutual-distillation as well. Blue and red denotes mutual-distillation models that are trained in the base and large group respectively. Models without colour are not co-trained with any other models. *Note that for base encoders, our results can slightly differ from numbers reported in (Gao et al., 2021) since different evaluation packages are used.

by approximately 4-5%.[9] Self-distillation usually brings an improvement of 2-3% (e.g., compare line 1.1, 1.2 with 1 in Tab. 1). Further, mutual-distillation brings another boost of 1-3% (e.g., compare line 3.3, 3.4 with 3.1, 3.2 in Tab. 1). Besides single-model results, the ensemble of mutual-distilled models are clearly better than the ensemble models of either (1) naively averaging the two initial SimCSE models' scores or (2) averaging predictions of two self-distilled models in a *post hoc* manner. This demonstrates the benefit of allowing models to communicate/synchronise with each other in all stages of the self-distillation training.

**Binary classification results (Tab. 2).** On QQP, QNLI, and MRPC, we observe similar trends as the STS tasks, and the improvement is sometimes even more significant (see full table in Appendix Tab. 11). We conjecture it is due to the fact that TRANS-ENCODER training also adapts models to the task domain (since tasks like QQP and QNLI are very different from Wikipedia data, which is where SimCSE models are trained on). We will discuss the domain adaptation effect in more details in analysis. Another point worth discussing is the benefit of mutual-distillation is more inconsistent on binary classification tasks, compared with STS tasks (again, see full table Tab. 11 in Appendix). We

---

[9]Similar trends observed on Mirror-BERT, see Appendix (App. §A.1).

| dataset→ | QQP | QNLI | MRPC | avg. |
|---|---|---|---|---|
| SimCSE-BERT-base | 80.38 | 71.38 | 75.02 | 75.59 |
| + TENC (bi) | 82.10 | 75.30 | 75.71 | 77.70 |
| + TENC (cross) | 82.10 | 75.61 | 76.21 | 77.97 |
| + TENC-mutual (bi) | 84.00 | 76.93 | 76.62 | 79.18 |
| + TENC-mutual (cross) | **84.29** | **77.11** | **77.77** | **79.72** |

Table 2: Binary classification task results. AUC scores are reported. We only demonstrate results for one model for brevity. Full table can be found in Appendix (Tab. 11).

| dataset→ | QQP | QNLI | MRPC | avg. |
|---|---|---|---|---|
| SimCSE-RoBERTa-base | 81.82 | 73.54 | 75.06 | 76.81 |
| + TENC (bi) | 82.56 | 71.67 | 74.24 | 76.16 |
| + TENC (cross) | 83.66 | 79.38 | 79.53 | 80.86 |
| + TENC-mutual (bi) | 81.71 | 72.78 | 75.51 | 76.67 |
| + TENC-mutual (cross) | **83.92** | **79.79** | **79.96** | **81.22** |

Table 3: A domain transfer setup: testing TRANS-ENCODER models trained with STS data directly on binary classification tasks. We only demonstrate results for one model for brevity. Full table can be found in Appendix (Tab. 12).

suspect it is due to the performance imbalance between the BERT and RoBERTa variants on these tasks (a significant performance gap exists in the first place).

**Domain transfer setup (Tab. 3).** One of the key questions we are keen to find out is how much of TRANS-ENCODER's success on in-domain tasks generalises/transfers to other tasks/domains. Specifically, does TRANS-ENCODER create universally better bi- and cross-encoders, or is it only task/domain-specific? To answer the question, we directly test models trained with the STS task data on binary classification tasks. For bi-encoders, i.e., TENC (bi), the results are inconsistent across setups. These results hint that training on in-domain data is important for optimal performance gains for unsupervised bi-encoders. This is in line with the finding of (Liu et al., 2021). However, for cross-encoders, surprisingly, we see extremely good transfer performance. W/ or w/o mutual-distillation, TENC (cross) models outperform the SimCSE baselines by large margins, despite the fact that tasks like QQP and QNLI are of a very different domain compared with STS. In fact, they are sometimes even better than models tuned on the in-domain task data (c.f. Tab. 11). This finding has deep implications. It hints that the cross-encoder architecture (cross-attention) is by design very suitable for sentence-pair modelling (i.e., strong inductive bias is already in the architecture design), and once its sentence-pair modelling capability is 'activated', it is an extremely powerful universal representation that can be transferred to other tasks. The reason that STS models outperform in-domain models in some tasks hint that more sentence-pairs as raw training data is welcome (since the STS train set is larger than the binary classification ones). As future work, we plan to explore mining sentence pairs for TRANS-ENCODER learning from a general corpus (such as Wikipedia). Both lexical- (such as BM25) and semantic-level (neural models such as bi-encoders) IR systems could be leveraged here.

## 5.2 DISCUSSIONS AND ANALYSIS

In this section we discuss some interesting phenomena we observed, justify design choices we made in an empirical way, and also show a more fine-grained understanding of what exactly happens when using TRANS-ENCODER.

**Initialise with the original weights or train sequentially? (Fig. 3a)** As mentioned in §2.2 and §2.3, we initialise bi-encoders with SimCSE/Mirror-BERT weights and cross-encoders with PLMs' weights (we call this strategy *refreshing*). We have also tried maintaining the weights of bi- and cross-encoders sequentially. I.e., we initialise the cross-encoder's weights with the bi-encoder which just created the pseudo labels and vice versa (we call this strategy *sequential*). The benefit of doing so is keeping all the legacy knowledge learned in the process of self-distillation in the weights. As

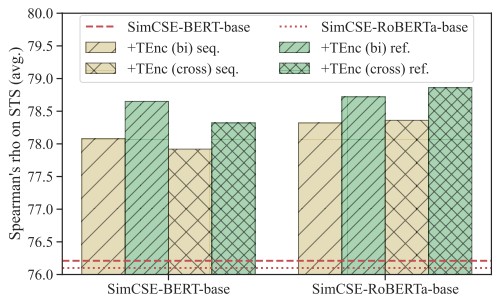 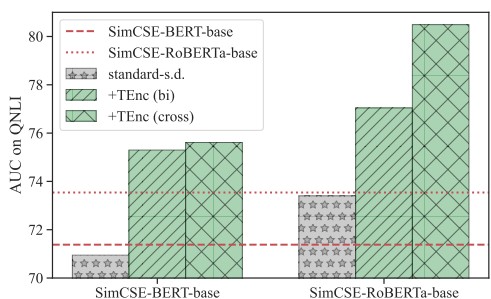

(a) Sequential vs. refreshing (on STS).      (b) TENC vs.standard self-distillation (on QNLI).

Figure 3: Ablation studies of TRANS-ENCODER regarding two design choices.

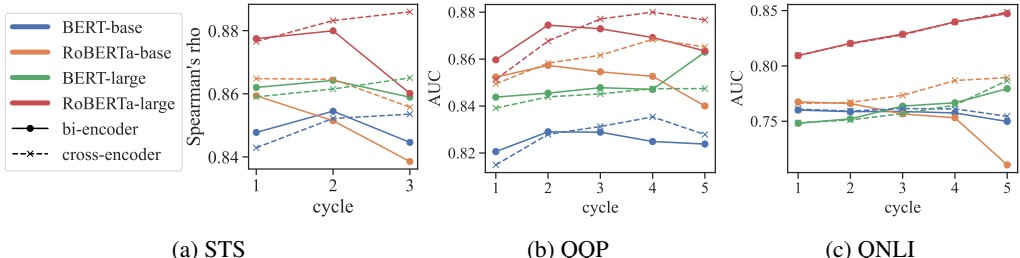

(a) STS           (b) QQP           (c) QNLI

Figure 4: TENC's performance against distillation cycles under different base models and tasks.

suggested in Fig. 3a (also in Appendix Tab. 13), *refreshing* is slightly better than *sequential*. We suspect it is because initialising with original weights alleviate catastrophic forgetting problem in self-supervised learning (SSL), similar to the moving average and stop-gradient strategy used in contrastive SSL methods such as MoCo (He et al., 2020).

**Self-distillation without alternating between different task formulations? (Fig. 3b)** If we disregard the bi- and cross-encoder alternating training paradigm but using the same bi-encoder architecture all along, the model is then similar to the standard self-distillation model (c.f. Figure 1 by Mobahi et al. (2020)). Theoretically, standard self-distillation could also have helped, especially considering that it adapts the model to in-domain data. However, as seen in Fig. 3b (and also Appendix Tab. 14), standard self-distillation lags behind TRANS-ENCODER by a significant amount, and sometimes underperforms the base model. Standard self-distillation essentially ensembles different checkpoints (i.e., the same model at different training phases provides different 'views' of the data). TRANS-ENCODER can be seen as a type of self-distillation model, but instead of using previous models to provide views of data, we use different task formulations (i.e., bi- and cross-encoder) to provide different views of the data. The fact that standard self-distillation underperforms TRANS-ENCODER by large margins suggests the effectiveness of our proposed bi- and cross-encoder training scheme.

**How many cycles is optimal? (Fig. 4)** Our approach iteratively bootstraps the performance of both bi- and cross-encoders. But how many iterations (cycles) are enough? We find that it heavily depends on the model and dataset, and could be unpredictable (since the 'optimality' depends on a relatively small dev set). In Fig. 4, we plot the TENC models' (self-distillation only) performances on dev sets in three tasks: STS, QQP, and QNLI. In general, the patterns are different across datasets and models.

**Does more training data help? (Fig. 5)** We control the number of training samples drawn and test the TRANS-ENCODER model (using SimCSE-BERT-base) on STS test sets. The average performance on all STS tasks are reported in Fig. 5, with three runs of different random seeds. There are in total 37,081 data points from STS. We only draw from these in-domain data until there is none left. It can be seen that for in-domain training, more data points are usually always beneficial. Next, we test with more samples drawn from another task, SNLI (Bowman et al., 2015), containing abundant sentence-pairs. The performance of the model further increases till 30k extra data points and starts to gradually decrease after that point. However, it is worth mentioning that there will be discrepancy of such trends across tasks and models. And when needed, one can mine as many sentence-pairs as desired from a general corpus (such as Wikipedia) for TRANS-ENCODER learning.

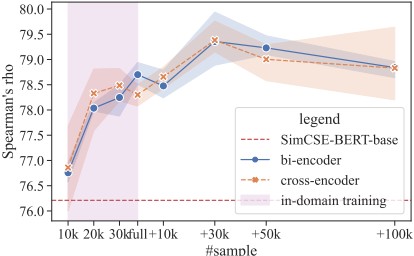

Figure 5: STS performance against number of training samples. 'full' means all STS in-domain data are used. After 'full', the samples are drawn from the SNLI dataset. Note that each point in the graph is an individual run.

## 6  RELATED WORK

Our work is closely related to (1) unsupervised sentence representation learning; (2) sentence-pair tasks using bi- and cross-encoders; and (3) self-distillation. Each has a large body of work which we can only provide a highly selected summary below.

**Unsupervised sentence representations.** Prior to the emergence of large PLMs such as BERT, popular unsupervised sentence embedding methods included Skip-Thoughts (Kiros et al., 2015), FastSent (Hill et al., 2016), and Quick-Thoughts (Logeswaran & Lee, 2018). They all exploit the co-occurrence statistics of sentences in large corpora (i.e., sentences under similar contexts have similar meanings). A recent paper DeCLUTR (Giorgi et al., 2021) follows such idea and formulate the training as a contrastive learning task. Very recently, there has been a growing interest in using individual raw sentences for self-supervised contrastive learning on top of PLMs. Contrastive Tension (Carlsson et al., 2021), Mirror-BERT (Liu et al., 2021), SimCSE (Gao et al., 2021), Self-Guided Contrastive Learning (Kim et al., 2021), ConSERT (Yan et al., 2021), BSL (Zhang et al., 2021), *inter alia*, all follow such an idea. Specifically, data augmentations on either input or feature space are used to create two views of the same sentence for contrastive learning. In our experiments, we use such methods as an initial step for creating a reasonably strong bi-encoder.

**Cross- and bi-encoder for pairwise sentence tasks.** Sentence-BERT (Reimers & Gurevych, 2019) points out the speed advantage of bi-encoders and proposes to train BERT with a siamese architecture for sentence-pair tasks. Poly-encoder (Humeau et al., 2020) proposes a hybrid architecture in which intermediate encodings are cached and a final layer of cross-attention is used to exploit inter-sentence interactions. Conceptually similar to our work, Di-pair (Chen et al., 2020), augmented Sentence-BERT (Thakur et al., 2021), RocketQA (Qu et al., 2021), and DvBERT (Cheng, 2021) distil knowledge in supervised cross-encoders to bi-encoders. We show that it is also possible to distil knowledge from bi- to cross-encoders.

**Self-distillation.** In standard self-distillation, models of the same architecture train on predictions by previous checkpoint(s) with the same task formulation (Furlanello et al., 2018; Mobahi et al., 2020). However, the standard approach does not alter the training paradigm (task formulation) as we do (bi- and cross-encoder training). In experiments, we have shown that alternating task formulation is the key to the success of our method.

## 7  CONCLUSION

We propose TRANS-ENCODER, an unsupervised approach of training bi- and cross-encoders for sentence-pair tasks. The core idea of TRANS-ENCODER is self-distillation in a smart way: alternatively training a bi-encoder and a cross-encoder (of the same architecture) with pseudo-labels created from the other. We also propose a mutual-distillation extension to mutually bootstrap two self-distillation models trained in parallel. On sentence-pair tasks including sentence similarity, question de-duplication, question-answering entailment, and paraphrase identification, we show strong empirical evidence verifying the effectiveness of TRANS-ENCODER. We also found the surprisingly strong generalisation capability of our trained cross-encoders across domains and tasks. Finally, we conduct thorough ablation studies and analysis to verify our design choices and shed insight on the model mechanism.

# 8 ACKNOWLEDGEMENTS

We thank Hanchen Wang, Giorgio Giannone, Arushi Goel for providing useful feedback during internal discussions.

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

| dataset→ | STS12 | STS13 | STS14 | STS15 | STS16 | STS-B | SICK-R | avg. |
|---|---|---|---|---|---|---|---|---|
| Mirror-RoBERTa-base | 64.67 | 81.53 | 73.05 | 79.78 | 78.16 | 78.36 | **70.03** | 75.08 |
| + TENC (bi) | 71.99 | 81.85 | 75.73 | 83.32 | **79.97** | 81.19 | 69.47 | 77.64 |
| + TENC (cross) | **73.10** | **82.31** | **76.49** | **83.71** | 79.64 | **81.70** | 69.70 | **78.09** |
| Mirror-RoBERTa-base-drophead | 68.15 | 82.55 | 73.47 | 82.26 | 79.44 | 79.63 | 71.58 | 76.72 |
| + TENC (bi) | 74.95 | 83.86 | 77.50 | 85.80 | 83.22 | 83.94 | 72.56 | 80.26 |
| + TENC (cross) | **75.70** | **84.58** | **78.35** | **86.49** | **83.96** | **84.26** | **72.76** | **80.87** |
| Contrastive-Tension-BERT-base | 61.98 | 76.82 | 68.35 | 77.38 | 76.56 | 73.99 | 69.20 | 72.04 |
| + TENC (bi) | 69.58 | **79.38** | **69.32** | **77.59** | 77.96 | 78.23 | **70.70** | 74.68 |
| + TENC (cross) | **70.67** | 79.27 | 69.26 | 76.77 | **78.78** | **79.08** | 70.66 | **74.93** |
| DeCLUTR-RoBERTa-base | 45.56 | 73.38 | 63.39 | 72.72 | 76.15 | 66.40 | 68.99 | 66.66 |
| + TENC (bi) | **62.40** | 77.55 | **70.05** | 80.28 | **81.81** | 77.31 | **72.60** | **74.57** |
| + TENC (cross) | 60.68 | **78.11** | 69.23 | **80.48** | 81.03 | **78.47** | 70.07 | 74.01 |

Table 4: English STS (models beyond SimCSE). Spearman's $\rho$ scores are reported.

| dataset→ | QQP | QNLI | MRPC | avg. |
|---|---|---|---|---|
| Mirror-RoBERTa-base | 78.89 | 73.73 | 75.44 | 76.02 |
| + TENC (bi) | **82.34** | 79.57 | 77.08 | 79.66 |
| + TENC (cross) | 82.00 | **79.87** | **78.40** | **80.09** |
| Mirror-RoBERTa-base-drophead | 78.36 | 75.56 | 77.18 | 77.03 |
| + TENC (bi) | 82.04 | 82.12 | 79.63 | 81.26 |
| + TENC (cross) | **82.90** | **82.52** | **81.38** | **82.27** |
| Contrastive-Tension-BERT-base | 78.95 | 69.73 | **72.86** | 73.85 |
| + TENC (bi) | 80.96 | 68.31 | 72.54 | 73.94 |
| + TENC (cross) | **81.22** | **74.17** | 71.33 | **75.57** |
| DeCLUTR-RoBERTa-base | 78.68 | 74.77 | 72.86 | 75.44 |
| + TENC (bi) | 83.64 | 82.87 | 74.55 | 80.35 |
| + TENC (cross) | **84.02** | **83.41** | **75.55** | **80.99** |

Table 5: Binary classification (models beyond SimCSE). AUC scores are reported.

# A   APPENDIX

## A.1   RESULTS WITH OTHER CONTRASTIVE LEARNING METHODS

In the main text, we experimented with SimCSE, but in theory TRANS-ENCODER is compatible with any unsupervised contrastive bi-encoder learning models. In this section, we experiment with Mirror-BERT (which is similar to SimCSE) and also two additional unsupervised contrastive models Contrastive-Tension (Carlsson et al., 2021) and DeCLUTR (Giorgi et al., 2021) to verify the robustness of TRANS-ENCODER.

As seen in Tab. 4 and Tab. 5, TRANS-ENCODER brings consistent and significant improvements to all base models, similar to what we have observed on SimCSE models. After TRANS-ENCODER training, the Mirror-BERT-based models perform even slightly better than SimCSE-based models, on both the STS and binary classification tasks. We suspect it is due to that SimCSE checkpoints are trained for more iterations (with the contrastive learning objective) and thus are more prone to overfit the training corpus.

## A.2   MORE DISCUSSIONS AND ANALYSIS

**How robust is TRANS-ENCODER? (Tab. 6)**   As mentioned, our main experiments used one fixed random seed. Due to the scale of the experiments, performing multiple runs with different random seeds for all setups is overly expensive. We pick STS and the base variants of BERT and RoBERTa

| dataset→ | mean | S.D. |
|---|---|---|
| SimCSE-BERT-base | | |
| + TENC (bi) | 78.72 | 0.16 |
| + TENC (cross) | 78.21 | 0.26 |
| SimCSE-RoBERTa-base | | |
| + TENC (bi) | 78.38 | 0.26 |
| + TENC (cross) | 78.90 | 0.34 |

Table 6: Mean and standard deviation (S.D.) of five runs on STS.

| Data→ | | STS | | SICK-R | |
|---|---|---|---|---|---|
| model→ | off-the-shelf | +TENC (bi) | +TENC (cross) | +TENC (bi) | +TENC (cross) |
| SimCSE-BERT-base | 72.22 | 71.84 | 71.16 | 74.13 | 74.43 |
| SimCSE-RoBERTa-base | 68.62 | 67.63 | 69.51 | 68.39 | 70.38 |
| SimCSE-BERT-large | 73.88 | 71.46 | 70.90 | 74.92 | 74.98 |
| SimCSE-RoBERTa-large | 71.23 | 68.36 | 67.90 | 72.63 | 73.13 |

Table 7: Compare TRANS-ENCODER models trained with all STS data (using STS-B's dev set) and SICK-R data only (using SICK-R's dev set). Large performance gains can be obtained when treating SICK-R as a standalone task.

as two examples for showing results of five runs (shown in Tab. 6). In general, the TRANS-ENCODER approach is quite robust and stable.

**STS and SICK-R as different tasks (Tab. 7, Fig. 6).** It is worth noting that in the STS main results table (Tab. 1), SICK-R is the only dataset where TRANS-ENCODER models lead to performance worse than the baselines in some settings (e.g., on SimCSE-BERT-large). We believe it is due to that SICK-R is essentially a different task from STS2012-2016 and STS-B since SICK-R is labelled with a different aim in mind. It focuses specifically on compositional distributional semantics.[10] To verify our claim, we train SICK-R as a standalone task using its official dev set instead of STS-B. All sentence pairs from SICK-R are used as raw training data. The results are shown in Tab. 7 and Fig. 6. As shown, around 3-4% gain can be obtained by switching to training on SICK-R only, confirming the different nature of the two tasks. This suggests that generalisation is also dependent on the standard of how relevance is defined between sequences in the target dataset, and training with an in-domain dev set maybe crucial when the domain shift between training and testing data is significant.

**Gradient derivation of BCE and MSE loss.** We show that BCE is essentially a normalised version of MSE where the gradient of $x_i$ is scaled by the sigmoid function ($\sigma(\cdot)$). Here are the gradients of the two loss functions:

$$\frac{\partial \mathcal{L}_{\text{BCE}}}{\partial x_i} = -\frac{1}{N}\Big(y_i(\sigma(x_i) - 1) + (1 - y_i)\sigma(x_i)\Big)$$
$$= -\frac{1}{N}\Big(\sigma(x_i) - y_i\Big) \qquad (4)$$
$$\frac{\partial \mathcal{L}_{\text{MSE}}}{\partial x_i} = -\frac{2}{N}\Big(x_i - y_i\Big).$$

As can be seen, the only difference lies in whether $x_i$ is scaled by $\sigma(\cdot)$. This gives a greater degree of freedom to $\mathcal{L}_{\text{BCE}}$ especially at the areas in the two ends (close to 0 or 1): a large change of $x_i$ results

---

[10]SICK-R "includes a large number of sentence pairs that are rich in the lexical, syntactic and semantic phenomena that Compositional Distributional Semantic Models (CDSMs) are expected to account for (e.g., contextual synonymy and other lexical variation phenomena, active/passive and other syntactic alternations, impact of negation, determiners and other grammatical elements), but do not require dealing with other aspects of existing sentential data sets (e.g., STS, RTE) that are not within the scope of compositional distributional semantics." (see http://marcobaroni.org/composes/sick.html).

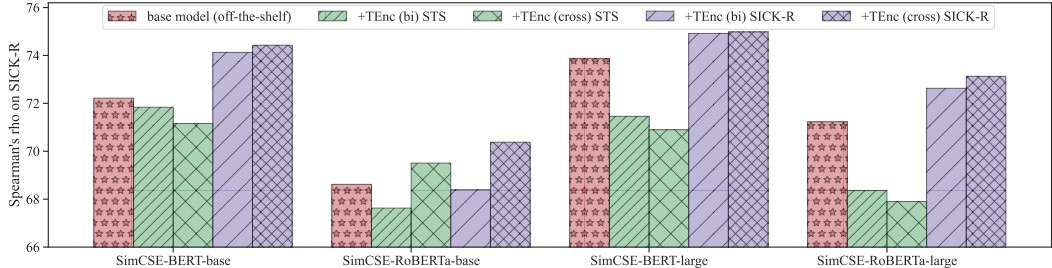

Figure 6: Graphical presentation of Tab. 7.

| $loss_{\text{bi-to-cross}}$ | $loss_{\text{cross-to-bi}}$ | result |
|---|---|---|
| BCE | MSE | TRANS-ENCODER configuration (✓) |
| BCE | BCE | cross-to-bi distillation won't converge (✗) |
| MSE | MSE | bi-to-cross distillation overfits completely to pseudo labels (✗) |
| MSE | BCE | bi-to-cross distillation overfits completely to pseudo labels (✗) & cross-to-bi distillation won't converge (✗) |

Table 8: Compare different loss function configurations.

in a small change of $\sigma(x_i)$, making the approximation of $y_i$ an easier task. In other words, in $\mathcal{L}_{\text{BCE}}$, it is in general easier for the model to push the gradient to $0$, while in $\mathcal{L}_{\text{MSE}}$, only a strict numerical match leads to $0$ gradient.

**Loss function configurations (Tab. 8).**  We experimented with loss function configurations comprehensively and found that it requires caution when choosing learning objectives for cross- to bi-encoder and bi- to cross-encoder distillation. As mentioned in the main text and the paragraph above, choosing MSE for bi-to-cross distillation causes severe overfitting to the pseudo scores and choosing BCE for cross-to-bi distillation fail to make the model converge. As a result, our configuration of using BCE and MSE for bi-to-cross and cross-to-bi distillations respectively becomes the only plausible solution (see Tab. 8).

For the regression loss used for cross-to-bi distillation, besides MSE, we also experimented with several other regression loss functions such as Log-Cosh but found no improvement.

**Do sentence pairs leak label information? (Tab. 9)**  TRANS-ENCODER has leveraged the sentence pairs given by the task. One concern is that the sentence pairs may be positively correlated and the model may have exploited this naïve bias. We highlight that this work does not have this concern as the tasks in our study test the *relative similarity ranking* of sentence pairs under the evaluation metrics (both spearman's rank used in STS and AUC used in binary classification are ranking-based). Therefore, the overall correlation distribution of sentence pairs does not reveal what is tested by the tasks in this study. To further verify the model has not learned label distribution from the task, we run a controlled setting where we treat all sentence pairs from the tasks as (pseudo-)positive pairs and apply a standard contrastive finetuning framework (i.e. Mirror-BERT and SimCSE finetuning) with Eq. (1) as the loss on these pairs. Performance improvement from this finetuning setup will indicate that the overall positive correlation from the sentence pairs can be exploited by the model. Results are shown in Tab. 9. Contrastive methods on pseudo-positive pairs almost always[11] underperform the original SimCSE and Mirror-BERT checkpoints significantly while TRANS-ENCODER models improve the checkpoints by large margins.

---

[11]The only exception is QNLI. We suspect it is because QNLI is from the QA domain and contrastive tuning on QNLI sentence pairs has some domain adaptation effect. According to Liu et al. (2021), contrastive tuning on randomly sampled sentences from the target domain can also achieve domain adaptation, and sentence pairs from the task are thus not a must.

| dataset→ | STS (avg.) | QQP | QNLI | MRPC | avg. |
|---|---|---|---|---|---|
| SimCSE-RoBERTa-base | 76.10 | 81.82 | 73.54 | 75.06 | 76.63 |
| + TENC (bi) | 78.36 | 84.13 | 77.08 | 75.29 | 78.72 |
| + TENC (cross) | **78.86** | **85.16** | **80.49** | **76.09** | **80.15** |
| + contrastive tuning | 73.93 | 74.00 | 77.56 | 75.25 | 75.19 |
| Mirror-RoBERTa-base-drophead | 76.72 | 78.36 | 75.56 | 77.18 | 77.00 |
| + TENC (bi) | 80.26 | 82.04 | 82.12 | 79.63 | 81.01 |
| + TENC (cross) | **80.87** | **82.90** | **82.52** | **81.38** | **82.27** |
| + contrastive tuning | 72.21 | 73.51 | 77.50 | 75.63 | 74.71 |

Table 9: Compare TRANS-ENCODER with SimCSE/Mirror-BERT-style contrastive tuning on sentence pairs given by the task.

| model size | dataset | # GPU required | time |
|---|---|---|---|
| TENC (base) | STS | 1 | ≈0.5 hrs |
| TENC (large) | STS | 1 | ≈1.5 hrs |
| TENC-mutual (base) | STS | 2 | ≈1.25 hrs |
| TENC-mutual (large) | STS | 2 | ≈3 hrs |
| TENC (base) | QQP | 1 | ≈1 hrs |
| TENC (large) | QQP | 1 | ≈2.5 hrs |
| TENC-mutual (base) | QQP | 2 | ≈2.25 hrs |
| TENC-mutual (large) | QQP | 2 | ≈5 hrs |
| TENC (base) | QNLI | 1 | ≈0.5 hrs |
| TENC (large) | QNLI | 1 | ≈1.75 hrs |
| TENC-mutual (base) | QNLI | 2 | ≈1.25 hrs |
| TENC-mutual (large) | QNLI | 2 | ≈3.5 hrs |
| TENC (base) | MRPC | 1 | ≈0.25 hrs |
| TENC (large) | MRPC | 1 | ≈0.75 hrs |
| TENC-mutual (base) | MRPC | 2 | ≈0.5 hrs |
| TENC-mutual (large) | MRPC | 2 | ≈1.5 hrs |

Table 10: Running time for all models across different datasets/tasks. We experimented with NVIDIA V100, A100 and 3090 and found the estimated time to be similar.

## A.3 TRAINING TIME

**Overall runtime.** We list the runtime of TRANS-ENCODER models on different tasks and configurations in Tab. 10. All base models can be trained under 2.5 hours and large models can be trained within 5 hours. TENC-mutual models usually take twice the amount of time training a self-distillation model. Also, TENC-mutual models require two GPUs while self-distillation models need only one. We have tested running TRANS-ENCODER on three types of Nvidia GPUs: V100 (16 GB), A100 (40 GB) and 3090 (24 GB). The runtimes are similar.

**Cross-encoder vs. bi-encoder.** For training one epoch, bi-encoder takes significantly less time than cross-encoders. However, bi-encoders take much longer time to converge (15 epochs vs. 3 epochs on binary classification tasks and 10 epochs vs. 1 epoch on STS; see Tab. 16). As a result, in practice, bi-to-cross and cross-to-bi distillations take similar amount of time.

## A.4 FULL TABLES

Here, we list the full tables of SimCSE binary classification results (Tab. 11); SimCSE domain transfer results (Tab. 12); and full tables for ablation studies (Tab. 13, Tab. 14), which are presented as figures in the main texts in Fig. 3.

| # dataset→ | QQP | QNLI | MRPC | avg. |
|---|---|---|---|---|
| *single-model results* | | | | |
| SimCSE-BERT-base | 80.38 | 71.38 | 75.02 | 75.59 |
| + TENC (bi) | 82.10 | 75.30 | 75.71 | 77.70 |
| + TENC (cross) | 82.10 | 75.61 | 76.21 | 77.97 |
| + TENC-mutual (bi) | 84.00 | 76.93 | 76.62 | 79.18 |
| + TENC-mutual (cross) | **84.29** | **77.11** | **77.77** | **79.72** |
| SimCSE-RoBERTa-base | 81.82 | 73.54 | 75.06 | 76.81 |
| + TENC (bi) | 84.13 | 77.08 | 75.29 | 78.83 |
| + TENC (cross) | **85.16** | **80.49** | 76.09 | **80.58** |
| + TENC-mutual (bi) | 84.36 | 77.29 | 76.90 | 79.52 |
| + TENC-mutual (cross) | 84.73 | 77.32 | **78.47** | 80.17 |
| SimCSE-BERT-large | 82.42 | 72.46 | 75.93 | 76.94 |
| + TENC (bi) | 84.53 | 78.22 | 78.42 | 80.39 |
| + TENC (cross) | 84.18 | 79.71 | 79.43 | 81.11 |
| + TENC-mutual (bi) | 85.72 | **81.61** | 79.59 | 82.31 |
| + TENC-mutual (cross) | **86.55** | 81.55 | **79.69** | **82.60** |
| SimCSE-RoBERTa-large | 82.99 | 75.76 | 77.24 | 78.66 |
| + TENC (bi) | 85.65 | 84.49 | 79.34 | 83.16 |
| + TENC (cross) | 86.16 | **84.69** | 81.00 | **83.95** |
| + TENC-mutual (bi) | 85.65 | 81.87 | 79.53 | 82.35 |
| + TENC-mutual (cross) | **86.38** | 83.14 | **81.77** | 83.76 |
| *ensemble results* (average predictions of two models) | | | | |
| SimCSE-base ensemble | 81.10 | 72.46 | 75.04 | 76.20 |
| TENC-base (bi) ensemble | 83.11 | 76.19 | 75.45 | 78.25 |
| TENC-base (cross) ensemble | 84.15 | **78.55** | **78.90** | **80.53** |
| TENC-mutual-base (bi) ensemble | 84.18 | 77.11 | 76.76 | 79.35 |
| TENC-mutual-base (cross) ensemble | **84.68** | 77.92 | 78.22 | 80.27 |
| SimCSE-large ensemble | 82.47 | 74.09 | 76.55 | 77.70 |
| TENC-large (bi) ensemble | 85.09 | 81.35 | 78.88 | 81.77 |
| TENC-large (cross) ensemble | 86.22 | **83.78** | 81.06 | **83.69** |
| TENC-mutual-large (bi) ensemble | 86.10 | 81.74 | 79.56 | 82.47 |
| TENC-mutual-large (cross) ensemble | **86.26** | 82.86 | **81.41** | 83.51 |

Table 11: Binary classification task results (SimCSE models; full table). AUC scores are reported.

## A.5 DATA STATISTICS

A complete listing of train/dev/test stats of all used datasets can be found in Tab. 15. Note that for STS 2012-2016, we dropped all sentence-pairs without a valid score. And the train sets include all sentence pairs (w/o labels) regardless of split in each task.

## A.6 HYPERPARAMETERS

The full listing of hyperparameters is shown in Tab. 16. Since performing a grid-search for all these hparams is ultra-expensive, we mainly used the default hparams from the sentence-BERT library.[12] The hparams for TENC and TENC-mutual are the same.

## A.7 PRE-TRAINED ENCODERS

A complete listing of URLs for all used PLMs is provided in Tab. 17.

---

[12]www.sbert.net

| dataset→ | QQP | QNLI | MRPC | avg. |
|---|---|---|---|---|
| SimCSE-BERT-base | 80.38 | 71.38 | 75.02 | 75.59 |
| + TENC (bi) | 80.80 | 72.35 | 74.53 | 75.89 |
| + TENC (cross) | 80.84 | **78.81** | **79.42** | 79.69 |
| + TENC-mutual (bi) | **83.24** | 72.88 | 75.86 | 77.33 |
| + TENC-mutual (cross) | 82.40 | 78.30 | 78.72 | **79.81** |
| SimCSE-RoBERTa-base | 81.82 | 73.54 | 75.06 | 76.81 |
| + TENC (bi) | 82.56 | 71.67 | 74.24 | 76.16 |
| + TENC (cross) | 83.66 | 79.38 | 79.53 | 80.86 |
| + TENC-mutual (bi) | 81.71 | 72.78 | 75.51 | 76.67 |
| + TENC-mutual (cross) | **83.92** | **79.79** | **79.96** | **81.22** |
| SimCSE-BERT-large | 82.42 | 72.46 | 75.93 | 76.94 |
| + TENC (bi) | 81.86 | 71.99 | 74.99 | 76.28 |
| + TENC (cross) | **83.31** | **79.62** | 79.93 | 80.95 |
| + TENC-mutual (bi) | 82.30 | 72.47 | 76.74 | 77.17 |
| + TENC-mutual (cross) | 83.04 | 79.58 | **81.18** | **81.27** |
| SimCSE-RoBERTa-large | 82.99 | 75.76 | 77.24 | 78.66 |
| + TENC (bi) | 82.34 | 70.88 | 76.05 | 76.42 |
| + TENC (cross) | 85.98 | 80.07 | 81.20 | 82.42 |
| + TENC-mutual (bi) | 85.28 | 71.56 | 76.81 | 77.88 |
| + TENC-mutual (cross) | **86.31** | **81.77** | **81.86** | **83.31** |

Table 12: Full table for domain transfer setup: testing TRANS-ENCODER (SimCSE models) trained with STS data directly on binary classification tasks.

| dataset→ | STS (avg.) |
|---|---|
| SimCSE-BERT-base | 76.21 |
| + TENC (bi) sequential | 78.08 |
| + TENC (cross) sequential | 77.92 |
| + TENC (bi) refreshing | **78.65** |
| + TENC (cross) refreshing | 78.32 |
| SimCSE-RoBERTa-base | 76.10 |
| + TENC (bi) sequential | 78.32 |
| + TENC (cross) sequential | 78.72 |
| + TENC (bi) refreshing | 78.36 |
| + TENC (cross) refreshing | **78.86** |

Table 13: Ablation: sequential training with the same set of weights vs. refreshing weights for all models.

| dataset→ | STS (avg.) | QNLI |
|---|---|---|
| SimCSE-BERT-base | 76.21 | 71.38 |
| + standard-self-distillation | 77.16 | 70.95 |
| + TENC (bi) | 78.65 | 75.30 |
| + TENC (cross) | 78.32 | 75.61 |
| SimCSE-RoBERTa-base | 76.10 | 73.54 |
| + standard-self-distillation | 76.25 | 73.40 |
| + TENC (bi) | 78.36 | 77.04 |
| + TENC (cross) | 78.86 | 80.49 |

Table 14: Ablation: compare TRANS-ENCODER with standard self-distillation.

| Dataset | |Train| | |Dev| | |Test| |
|---|---|---|---|
| STS 2012 | - | - | 3,108 |
| STS 2013 | - | - | 1,500 |
| STS 2014 | - | - | 3,750 |
| STS 2015 | - | - | 3,000 |
| STS 2016 | - | - | 1,186 |
| STS-B | - | 1,500 | 1,379 |
| SICK-R | - | 495 | 4,906 |
| STS train (full)⋆ | 37,081 | - | - |
| QQP | 21,000 | 1,000 | 10,000 |
| QNLI | 15,463 | 1,000 | 4,463 |
| MRPC | 5,801 | 408 | 1,725 |

Table 15: A listing of train/dev/test stats of all used datasets. ⋆: a collection of all individual sentence-pairs from all STS tasks.

| task | direction | learning rate | batch size | epoch | max token length | cycles |
|---|---|---|---|---|---|---|
| *base models* | | | | | | |
| STS | bi → cross | 2e-5 | 32 | 1 | 64 | 3 |
| STS | cross → bi | 5e-5 | 128 | 10 | 32 | 3 |
| binary | bi → cross | 2e-5 | 32 | 3 | 64 | 5 |
| binary | cross → bi | 5e-5 | 128 | 15 | 32 | 5 |
| *large models* | | | | | | |
| STS | bi → cross | 2e-5 | 32 | 1 | 64 | 3 |
| STS | cross → bi | 5e-5 | 64 | 10 | 32 | 3 |
| binary | bi → cross | 2e-5 | 32 | 3 | 64 | 5 |
| binary | cross → bi | 5e-5 | 64 | 15 | 32 | 5 |

Table 16: A listing of hyperpamters used for all TRANS-ENCODER models.

| model | URL |
|---|---|
| BERT | https://huggingface.co/bert-base-uncased |
| RoBERTa | https://huggingface.co/roberta-base |
| SimCSE-BERT-base | https://huggingface.co/princeton-nlp/unsup-simcse-bert-base-uncased |
| SimCSE-RoBERTa-base | https://huggingface.co/princeton-nlp/unsup-simcse-roberta-base |
| SimCSE-BERT-large | https://huggingface.co/princeton-nlp/unsup-simcse-bert-large-uncased |
| SimCSE-RoBERTa-large | https://huggingface.co/princeton-nlp/unsup-simcse-roberta-large |
| Mirror-RoBERTa-base | https://huggingface.co/cambridgeltl/mirror-roberta-base-sentence |
| Mirror-RoBERTa-base-drophead | https://huggingface.co/cambridgeltl/mirror-roberta-base-sentence-drophead |
| Contrastive-Tension-BERT-base | https://huggingface.co/Contrastive-Tension/BERT-Base-CT |
| DeCLUTR-RoBERTa-base | https://huggingface.co/johngiorgi/declutr-base |

Table 17: A listing of HuggingFace URLs of all PLMs used in this work.

