# OpenReview forum: "Trans-Encoder: Unsupervised sentence-pair modelling through self- and mutual-distillations"
_ICLR.cc/2022/Conference — ICLR 2022 Poster_

### Official Review · Reviewer_DZEu · 2021-10-30

**Correctness:** 4
**Technical Novelty And Significance:** 3
**Empirical Novelty And Significance:** 3
**Recommendation:** 6
**Confidence:** 4

**Main Review:**

Strengths
- The author experimented on stronger models (RoBERTa-large) too, and the results still hold.
- The writing is very clear. The figures are great. Reproducibility is likely strong.

Concerns/comments
- The paper will be stronger if the authors can include more hypotheses or justification on why, in some cases (see Table 1 and Table 8), TNec-mutual works much better than TEnc (cross), especially because the bi-encoders do not perform as well.
- It will be great if the paper includes more details on the training time. How do the training time compare among different approaches, for GLUE tasks (especially larger ones)? How difficult is it to train the mutual distillation?
- The mutual distillation approach requires doing self-distillation on multiple pretrained language models. What’s the aggregate training overhead of mutual distillation vs. cross-encoder only vs. bi-encoder only?

More questions...
- What's the training stability for self-distillation? Will results have a small standard deviation?

Minor
- I’m interested in seeing the performance on more difficult tasks that might require more world knowledge (e.g., MNLI), but this is not necessary.
- Typo on page 2 “distil” -> “distill”
- Page 3, paragraph 3, should be Liu et al. (2021) instead of \citep?


**Summary Of The Paper:**

This paper deals with tasks where we need to do pairwise comparison between two sequences. Specifically, the paper experimented on STS, QQP, QNLI, MRPC. The paper proposes integrating two models together: bi-encoders and cross-encoders.

A bi-encoder encodes each of the two sequences separately and maps to a common embedding space. A cross-encoder first concatenates two sequences, and the concatenation is sent to the model. There’s more computation for the cross-encoder case, but the cross-encoder usually outperforms the bi-encoder.

Figure 1 illustrates the authors’ proposal well. First, start with a pretrained language model. Then, the bi-encoder generates pseudo-labels for the cross-encoder to learn. Next, the cross-encoder generates (more accurate) pseudo-labels for the bi-encdoer to learn. We reiterate this process. This approach is named self-distillation.

The author proposes an extension called mutual distillation where the authors do self-distillation on multiple pretrained language models in parallel (but the paper only experiments with two--BERT and RoBERTa).

On STS, QQP, QNLI, MRPC, the mutual distillation approach archives good performances (much better than the bi-encoder baseline SimCSE).


**Summary Of The Review:**

The approach is straightforward and nicely tied the bi-encdoer approach with the cross-encoder approach. The improvements compared to SimCSE is significant. The writing is clear and the reproducibility is good. However, I would expect more theoretical justification on why TEnc-mutual could be much higher than TEnc-cross. Moreover, training stability (of self distillation) and aggregate training time (for mutual distillation) deserve more discussion in my opinion.

---

> ### Author Response · Authors · 2021-11-22
> **response to Reviewer DZEu**
>
> *Reviewer DZEu*: **...The paper will be stronger if the authors can include more hypotheses or justification on why, in some cases (see Table 1 and Table 8), TNec-mutual works much better than TEnc (cross), especially because the bi-encoders do not perform as well...**
>
> TEnc-mutual models are usually always stronger since they use mutual-distillation and essentially have ensembled information from multiple (two) pretrained language models (PLMs). TEnc (cross) does not have mutual-distillation and uses only one PLM and is thus usually always worse. We previously mentioned model acronyms in Section 2.4 and the caption of Table 1 but we have added more explanation about how models are named to avoid misinterpretation.
>
> *Reviewer DZEu*: **...It will be great if the paper includes more details on the training time...**
>
> Thanks for pointing out the missing training overhead. We have added a comprehensive section to discuss the runtime of all models under all configurations, and also the runtime of cross-encoder vs. bi-encoder. Please see Appendix A.3 and Table 10 for details.
> As a summary, all base models can be trained under 2.5 hours and large models can be trained within 5 hours. TEnc-mutual models usually take twice the amount of time training a self-distillation model. Also, TEnc-mutual models require two GPUs while self-distillation models need only one. We have tested running Trans-Encoder on three types of Nvidia GPUs: V100 (16 GB), A100 (40 GB) and 3090 (24 GB). The runtimes are similar.
>
> Cross-encoder vs. bi-encoder: For training one epoch, bi-encoder takes significantly less time than cross-encoders. However, bi-encoders take much longer times to converge (15 epochs vs. 3 epochs on binary classification tasks and 10 epochs vs. 1 epoch on STS). As a result, in practice, bi-to-cross and cross-to-bi distillations take a similar amount of time.
>
> *Reviewer DZEu*: **"...What's the training stability for self-distillation? Will results have a small standard deviation?..."**
>
> We had a training stability analysis in Appendix A.2 *“How robust is Trans-Encoder?”*, in which we experimented with 5 different random seeds on two base models and found the standard deviation on the STS tasks (avg.) to be between 0.16 and 0.34. Trans-Encoder training is generally quite stable. Please also see Table 6 for details.

---

> > ### Comment · Reviewer_DZEu · 2021-11-30
> > **Thank you authors**
> >
> > - I reviewed Table 10 and Appendix 3. The training time is reasonable.
> > - Moreover, the training stability (using different random seeds) is encouraging.
> > - The revised appendix A.2 (comparing between BCE and MSE losses) is interesting.

---

> > > ### Author Response · Authors · 2021-11-30
> > > **thank you for reading our response!**
> > >
> > > thank you for reading our response!

---

### Official Review · Reviewer_53d8 · 2021-11-02

**Correctness:** 2
**Technical Novelty And Significance:** 3
**Empirical Novelty And Significance:** 2
**Recommendation:** 5
**Confidence:** 3

**Main Review:**

Strengths:
1. TRANS-ENCODER is the first completely unsupervised cross-encoder for sentence similarity.

2. TRANS-ENCODER outperforms state-of-the-art unsupervised sentence encoders on the sentence similarity benchmarks.

Weaknesses:
1. It is counter-intuitive that in the "bi- to cross-encoder" phase, the learnt cross-encoder outperforms the bi-encoder which produces the labeled data for the cross-encoder. It is also surprising that the student bi-encoder sometimes outperforms its teacher cross-encoder in the "cross- to bi-encoder" phase. The paper lacks the theoretical analysis and empirical investigation of why exactly these happened to TRANS-ENCODER.

2. Although the labels are not used, the test sets of any dataset should not be used for training. They should be considered unseen and held-out in the training phase.

3. What is the rationale for using different training losses for "bi- to cross-encoder" and "cross- to bi-encoder" (soft BCE vs. MSE)? Is it possible to use the same loss for the two phases?

**Summary Of The Paper:**

This paper proposes TRANS-ENCODER, an unsupervised approach to training bi-encoders and cross-encoders for sentence similarity tasks such as information retrieval, natural language inference, semantic textual similarity and clustering. The core idea of TRANS-ENCODER is self-distillation that alternatively trains a bi-encoder and a cross-encoder with pseudo-labels created from the other. The authors also propose a mutual-distillation extension to mutually bootstrap two self-distillation models trained in parallel. The effectiveness of TRANS-ENCODER is verified by empirical evidence.

**Summary Of The Review:**

Overall, this paper introduces an unsupervised method that seems to effective in sentence similarity tasks, but it lacks the insights and empirical investigation of why this method works well. The work is not well-supported. So I would consider it marginally below the acceptance threshold.

---

> ### Author Response · Authors · 2021-11-22
> **response to Reviewer 53d8**
>
> *Reviewer 53d8*:  **...It is also surprising that the student bi-encoder sometimes outperforms its teacher cross-encoder in the "cross- to bi-encoder" phase...**, **...
> What is the rationale for using different training losses for "bi- to cross-encoder" and "cross- to bi-encoder" (soft BCE vs. MSE)? Is it possible to use the same loss for the two phases?
> ...**
>
> We believe this has to do with our observation that “cross-encoders are more easy to overfit”. Since cross-encoders are strong function approximators, they can easily overfit to the pseudo scores given by bi-encoders and lose generalization capability, ending up having lower scores than the bi-encoders. This is also why we chose different loss functions for bi-to-cross and cross-to-bi distillations.
> We answer this together with the “BCE vs. MSE” concern as these two questions are coupled intrinsically. In Section 2.3, we have added a new paragraph to more thoroughly discuss this:
> *“Choice of loss functions: maintaining student-teacher discrepancy is the key. Intuitively, MSE allows for more accurate distillations since it punishes any numerical discrepancies between predicted scores and labels on the instance level. However, in practice, we found that using MSE for bi-to-cross distillation aggravates the overfitting issue for cross-encoder: the cross-encoder, with its strong sentence-pair modelling capability, completely overfits to the mapping between concatenated sentence pairs and the pseudo scores. This elimination of discrepancy between model predictions and pseudo labels harms generalisation and the iterative learning cycle cannot continue (as the predicted scores by the student model will be the same as the teacher model). We thus use BCE loss for bi-to-cross since BCE is a more forgiving loss function compared with MSE. According to our gradient derivation (App.§A.2), BCE essentially is a temperature-sharpened version of MSE, which is more tolerant towards numerical discrepancies. This prevents cross-encoders from overfitting to the pseudo labels completely. Similar issue does not exist for the cross-to-bi distillation as for bi-encoders, two input sequences are separately encoded and the model does not easily overfit to the labels. We have a more thorough discussion in App. §A.2 and Tab. 8 to highlight the rationales of the current configuration..”*
>
> When we designed the model, we experimented with loss function configurations comprehensively and found that it requires caution when choosing learning objectives for cross- to bi-encoder and bi- to cross-encoder distillation. As mentioned above, choosing MSE for bi-to-cross distillation causes severe overfitting to the pseudo scores and choosing BCE for cross-to-bi distillation fails to make the model converge. As a result, our configuration of using BCE and MSE for bi-to-cross and cross-to-bi distillations respectively becomes the only plausible solution. Please also see Table 8 in Appendix for a clear comparison. We have added this discussion to the paper as we realised that this is an important observation for readers to understand the inner working mechanism of Trans-Encoder.
>
> *Reviewer 53d8*: **...Although the labels are not used, the test sets of any dataset should not be used for training...**
>
> We think using all sentence pairs from the dataset is a realistic setup for sentence-pair tasks as in these tasks sentence pairs are always given, even at test time (for some STS datasets, only test pairs exist). Since they are part of the input data, they can be used to refine the model before outputting the prediction. As a result, our approach is a task-specific “unsupervised sentence-pair modelling” method, which is what we claim. Indeed, Trans-Encoder is not an “unsupervised generic sentence embedding” method. We will reiterate this in the paper!
> Similar practices exist in tasks such as machine translation and QA where source language test data or unlabelled questions are used for ad-hoc processing (Sutskever et al, ACL 2015) or back-translation augmentation (Huck et al, ACL 2019).
>
> Sutskever et al., ACL 2015: Addressing the Rare Word Problem in Neural Machine Translation https://aclanthology.org/P15-1002.pdf
> Huck et al, ACL 2019: Better OOV Translation with Bilingual Terminology Mining https://aclanthology.org/P19-1581.pdf

---

### Official Review · Reviewer_ofUo · 2021-11-02

**Correctness:** 4
**Technical Novelty And Significance:** 2
**Empirical Novelty And Significance:** 3
**Recommendation:** 6
**Confidence:** 4

**Main Review:**

# Strengths (Reasons to Accept)
- Solid work. What is claimed by the paper is clear and persuasive.
- This paper points out the convention in the sentence-pair modeling literature that bi- and cross-encoders have been considered separately. From this, it proposes an intuitive combination of the two encoder paradigms.
- When sticking to the experimental environment reported in the paper, the improvement in performance looks significant.

# Weaknesses (Reasons to reject)
- It is hard to say that the proposed method is novel, especially considering that all of the technical components used in this paper (e.g., pre-trained language models, contrastive learning for sentence modeling, knowledge distillation between two similar models, pseudo labeling for data augmentation/generation) are quite common in the literature. I believe that the main novelty of this paper only comes from the fact that it proposes for the first time (to the best of my knowledge) to combine bi- and cross-encoders. However, the specific way introduced to merge the encoders is not that special.
- Sentence modeling using (fine-tuned) pre-trained language models is one of the most active research areas in NLP. Considering this, it is questionable whether the proposed experiments are comprehensive enough to back up the paper's claim. Even though there exist so many other concurrent and similar works for sentence modeling other than SimCSE, this paper only focuses on incrementally refining SimCSE. To claim that the proposed method is universally effective, it would be desired to test the method on top of other similar sentence modeling frameworks in addition to SimCSE.
- After all, it is still unclear why the iterative and coupled fine-tuning of bi- and cross-encoders results in performance improvement. Assuming that the information contained in the same dataset is ideally equal, what is the specific factor that can be easily captured by cross-encoders but not by bi-encoders (and vice versa)? The reported improvement in performance is entirely due to the combination of two different encoder paradigms? How about using two separate bi-encoders or cross-encoders? What is the main and independent factor that contributes to the improvement most?


**Summary Of The Paper:**

# Summary

- This paper highlights the fact that bi-encoders (sentence encoders) and cross-encoders (sentence-pair encoders) have been considered somewhat separately in the literature of sentence-pair modeling, and that cyclic knowledge distillation from one to another (and vice versa) can be effective for improving the performance of both the encoder stereotypes.
- The authors combine the two sentence modeling paradigms into an iterative joint framework, which is called TRANS-ENCODER, to simultaneously learn enhanced bi- and cross-encoders.
- Specifically, given the original SimCSE or Mirror-BERT checkpoints provided by prior work, the proposed framework first (pseudo-)labels sentence pairs sampled from the task of interest by utilizing the off-the-shelf sentence encoders. And then, the labeled sentence pairs are utilized to improve the quality of cross-encoders. Further, the fine-tuned cross-encoders can also be exploited to label other sentence pairs which are again able to be used as a dataset for tuning the bi-encoders (sentence encoders). The framework repeats this process until both the bi- and cross-encoders become satisfactory in terms of their performance.
- The proposed method is evaluated on semantic textual similarity (STS) tasks and binary classification datasets such as QQP, QNLI, and MRPC. The results demonstrate that the method improves both the bi- and cross-encoders to be more suitable for sentence-pair modeling tasks.

**Summary Of The Review:**

- This paper clearly points out that bi- and cross-encoders have been evaluated and dealt with separately in the literature. From this, it proposes to combine the strengths of the two different paradigms by relying on cyclic knowledge distillation (or data augmentation).
- The paper is generally well-written and easy to understand. However, it is a little bit doubtful whether (1) the proposed method is novel enough and (2) the reported experiments are comprehensive enough. Furthermore, I feel that this paper does not contain a thorough analysis on "why" the proposed method works, which is what I want to know the most.
- Therefore, my suggstion is a weak accept (or on the borderline to be accepted).

---

> ### Author Response · Authors · 2021-11-22
> **response to Reviewer ofUo**
>
> *Reviewer ofUo*: **...It is hard to say that the proposed method is novel...**
>
> We believe Trans-Encoder is novel and worth being seen by the community for a few reasons:
> 1. Trans-Encoder sets a clear new SotA for the very competitive STS task by a very significant margin, suggesting this is an extremely effective method and would interest a wide range of readers.
> 2. While cross-encoders have been the predominant sentence-pair modelling technique in supervised tasks, it is the very first approach to train completely unsupervised cross-encoders for sentence-pair modelling.
> 3. The way how different components are combined is new: the design of the framework is non-trivial and needs insights since there are many unique challenges in the self-distillation process between the bi-encoders and cross-encoders (e.g. the bi-to-cross process is more likely to overfit and therefore requires a careful choice of loss functions; model weights need to be refreshed for every step instead of being used sequentially, etc.)
> 4. Unsupervised sentence modelling is a hard and emerging research direction. Trans-Encoder advances this research direction by presenting a practical and effective technical solution (accompanied by ablation studies on what training configurations work and why for future guidance) that develops the potential of several key machine learning concepts (including contrastive learning and knowledge distillation) one step further in sentence modelling.
>
> *Reviewer ofUo*: **...to claim that the proposed method is universally effective, it would be desired to test the method on top of other similar sentence modeling frameworks in addition to SimCSE...**
>
> We agree that verifying Trans-Encoder on models beyond SimCSE is important. For the original version of the paper, we also included results on two variants of Mirror-BERT (Liu et al., EMNLP 2021) in the appendix. We mainly considered SimCSE and Mirror-BERT as they report the previous SotA performance on STS. For the updated version of the paper, we have also added results for DeCLUTR (Giorgi et al., ACL 2021) and Contrastive-Tension (Carlsson et al., ICLR 2021). Please see Appendix A.1 and Table 4 & 5 for more details. In a nutshell, the trend is quite consistent across different contrastive learning methods, suggesting the robustness of Trans-Encoder.
>
> *Reviewer ofUo*: ***...what is the specific factor that can be easily captured by cross-encoders but not by bi-encoders (and vice versa)? How about using two separate bi-encoders or cross-encoders?...***
>
> - We believe the unique strengths of bi- and cross-encoders are: bi-encoder is usually more prone to overfitting and cross-encoder is the better task formulation for sentence-pair modelling. We had a discussion about why cross-encoders can capture better semantic similarity in Section 2.2:
> *“The cross-encoder directly discovers the similarity between two sentences through its attention heads, finding more accurate cues to justify the relevance score. The ability of discovering such cues could then generalise to unseen data, resulting in stronger sentence-pair scoring capability than the original bi-encoder. From a knowledge distillation perspective, we can view the bi- and cross-encoder as the teacher and student respectively. In this case the student outperforms the teacher, not because of stronger model capacity, but smarter task formulation.”*
>
> - We added a discussion about why bi-encoders are less likely to overfit at the end of Section 2.3.
> The most crucial independent factor that contributes to the success of Trans-Encoder is the bi- and cross-encoder alternation in the self-distillation process.  Regarding using two separate bi-encoders, we empirically tested this idea and added the results to Section 5.2:
> *“If we disregard the bi- and cross-encoder alternating training paradigm but using the same bi-encoder architecture all along (but different initializations), the model is then similar to the standard self-distillation model. Theoretically, standard self-distillation could also have helped. However, as seen in FIg. 3b standard self-distillation lags behind Trans-Encoder by a significant amount, and sometimes underperforms the base model. Standard self-distillation essentially ensembles different checkpoints. Trans-Encoder can be seen as a type of self-distillation model, but instead of using previous models to provide views of data, we use different task formulations (i.e., bi- and cross-encoder) to provide different views of the data. The fact that standard self-distillation underperforms Trans-Encoder by large margins suggests the effectiveness of our proposed bi- and cross-encoder training scheme. ”*
>
> Through this experiment we have verified that the alternation between training schemes is essential. Note that we cannot use two cross-encoders to learn unsupervised representations since cross-encoders need to be trained by pseudo-scores generated by bi-encoders.

---

### Official Review · Reviewer_z3e4 · 2021-11-03

**Correctness:** 3
**Technical Novelty And Significance:** 3
**Empirical Novelty And Significance:** 3
**Recommendation:** 5
**Confidence:** 4

**Main Review:**

S1. The proposed self-distillation framework is interesting and technically sound. The joint training of bi-encoder and cross-encoder goes beyond the existing self-distillation framework, which I believe is the major novelty of this paper. After going through the presented methods in this paper, I'm quite positive that improvements got to be achieved over baselines like SimCSE.

S2. Comprehensive experiments are performed, which clarify most of the critical aspects about the proposed methods.

W1. My biggest concern. I'm afraid that the proposed framework is not really unsupervised (I would call it pseudo unsupervised instead). In fact, it relies on labeled data: although labels are discarded, the pairwise relationships of the sentences are preserved, in which a large portion of the sentence pairs are positively correlated. The paper has to demonstrate that it may still work on a plain corpus where no such paired data is available; otherwise, it will be unrealistic as an unsupervised method.

W2. A minor concern towards the experiment. In table 1, TENC (cross) underperforms TENC (bi) in many cases, which is counter-intuitive, and kind of goes against the statement made in section 2.2.

**Summary Of The Paper:**

This paper focuses on the unsupervised training of bi-encoder and cross-encoder for sentences. A self-distillation framework is proposed, in which the bi-encoder and cross-encoder may iteratively distill knowledge from each other. The experiments are performed with typical sentence pair datasets like STS, QQP, QNLI, MRPC, where improvements are achieved over SimCSE and mirror-BERT.

**Summary Of The Review:**

The paper is interesting and technically sound. The experiments are comprehensive and inspiring. However, the assumption of pair sentences will probably make it unrealistic in reality.

---

> ### Author Response · Authors · 2021-11-22
> **response to the Reviewer z3e4**
>
> *Reviewer z3e4*: **...W1. My biggest concern. I'm afraid that the proposed framework is not really unsupervised (I would call it pseudo unsupervised instead)...**
>
> We agree that Trans-Encoder is not an *“unsupervised universal sentence embedding”* method but an *“unsupervised sentence-pair modelling”* method (which is what we claim, we will reiterate this clearer in the paper). In sentence-pair tasks such as QNLI, sentence pairs are always provided and used as input to the model. Trans-Encoder can then be used in this realistic setting where the model predicts similarity scores or binary classification labels on the given sequence pairs in a specific task (e.g. whether a sentence answers the question in QNLI).
>
> Regarding whether the overall distribution of sentence pair relationships in the task can leak label information, we have added a paragraph to discuss this more thoroughly (please see *“Do sentence pairs leak label information?”* and Table 9 in Appendix A.2 ):
>
> *“Trans-Encoder has leveraged the sentence pairs given by the task. One concern is that the sentence pairs may be positively correlated and the model may have exploited this naive bias. We highlight that our study does not have this concern as the tasks in our study test the “relative similarity ranking” of sentence pairs under the evaluation metrics (both spearman's rank used in STS and AUC used in binary classification are ranking-based). Therefore, the overall correlation distribution of sentence pairs does not reveal what is tested by the tasks in this study.*
>
> *To further verify the model has not learned label distribution from the task, we run a controlled setting where we treat all sentence pairs from the tasks as (pseudo-)positive pairs and apply a standard contrastive finetuning framework (i.e. Mirror-BERT and SimCSE finetuning) with infoNCE as the loss on these pairs. Performance improvement from this finetuning setup will indicate that the overall positive correlation from the sentence pairs can be exploited by the model. Results are shown in Table 9. Contrastive methods on pseudo-positive pairs almost always underperform the original SimCSE and Mirror-BERT checkpoints significantly while Trans-Encoder models improve the checkpoints by large margins.”*
>
> In an early exploration of this work, we tried using BM25 and also unsupervised bi-encoders to mine sentence pairs from a raw corpus for Trans-Encoder learning and observed consistent but weaker improvement. However, we found that the sampling strategy is extremely important and models can be extremely sensitive to the strategy used. This is an underexplored direction and requires substantial work to thoroughly experiment and discuss. This is why we mentioned in the paper that *“As future work, we plan to explore mining sentence pairs for Trans-Encoder learning from a general corpus (such as Wikipedia). Both lexical- (such as BM25) and semantic-level (neural models such as bi-encoders) IR systems could be leveraged here.”*
>
> *Reviewer z3e4*: **...W2. A minor concern towards the experiment. In table 1, TENC (cross) underperforms TENC (bi) in many cases, which is counter-intuitive...**
>
> It is true that in supervised settings, cross-encoders almost always outperform bi-encoders (we have modified the paper to specify “in supervised settings”). But for an unsupervised setup, it is a bit more complicated. If we take an average of all the numbers from our paper, the overall trend is still that cross-encoders outperform bi-encoders. But indeed in many specific cases, it’s the opposite. The reason is that cross-encoders are more likely to overfit compared with bi-encoders: cross-encoders take concatenated sentence pairs as input and learn a mapping between the pair to the label while bi-encoders encode the two sequences separately. As a result, it is harder for bi-encoders to learn the task and therefore overfit to the pseudo scores than cross-encoders. This leads to the bi-encoders’ better generalisation in some cases.
>
> We wrote a new paragraph in the main text to further discuss why cross-encoders overfit more easily. In particular, we performed controlled experiments to compare cross-encoders and bi-encoders with different loss functions, and we show that the cross-encoders are less robust to loss configurations than bi-encoders. For details please see *“Choice of loss functions: maintaining student-teacher discrepancy is the key.”* at the end of Section 2.3. or the general response.

---

### Author Response · Authors · 2021-11-22
**General response**

We thank all four reviewers for their comments and suggestions. They have helped us to improve the paper substantially.

Based on some shared comments, we’d like to reiterate two points (also mentioned in the specific responses) that might be interesting to all reviewers:

***A. Why is Trans-Encoder novel and worth being seen by the community?***
1. Trans-Encoder sets a clear new SotA for the very competitive STS task by very significant margin, suggesting this is a highly effective method that would be interesting for a wide range of readers.
2. While cross-encoders have been the predominant sentence-pair modelling technique in supervised tasks, it is the very first approach to train completely unsupervised cross-encoders for sentence-pair modelling.
3. The way how different components are combined is new: the design of the framework is non-trivial and needs insights since there are many unique challenges in the self-distillation process between the bi-encoders and cross-encoders. For example, the bi-to-cross process is more likely to overfit and therefore requires a careful choice of loss functions; model weights need to be refreshed for every step instead of being used sequentially, etc.
4. Unsupervised sentence modelling is a challenging and emerging research direction. Trans-Encoder advances this research direction by presenting a practical and effective solution (accompanied by ablation studies on what training configurations work and why for future guidance). This method develops the potential of several key machine learning concepts (including contrastive learning and knowledge distillation) one step further in the sentence modelling domain.

***B. Why cross-encoders sometimes underperform bi-encoders in our unsupervised setup?***
- If we take an average of all the numbers from our paper, the overall trend is still that cross-encoders outperform bi-encoders. But indeed, in many specific cases it’s the opposite. The reason is that cross-encoders are more likely to overfit compared with bi-encoders: cross-encoders take concatenated sentence pairs as input and learn a mapping between the pair to the label while bi-encoders encode the two sequences separately. As a result, it is harder for bi-encoders to learn the task and therefore overfit to the pseudo scores than cross-encoders. This leads to the bi-encoders’ better generalisation in some cases.

- We wrote a new paragraph in the main text to further discuss why cross-encoders overfit more easily. In particular, we performed controlled experiments to compare cross-encoders and bi-encoders with different loss functions (table below), and we showed that the cross-encoders are less robust to loss configurations than bi-encoders.  Please see *“Choice of loss functions: maintaining student-teacher discrepancy is the key.”* at the end of Section 2.3.

|loss_bi_to_cross   | loss_cross_to_bi   | result |
|---|---|---|
| BCE  | MSE  | trans-encoder configuration   |
| BCE  | BCE  | cross-to-bi distillation won't converge  |
| MSE  | MSE  | bi-to-cross distillation overfits to pseudo labels  |
| MSE  | BCE  |  bi-to-cross distillation overfits to pseudo labels & cross-to-bi distillation won't converge  |

- Note that unsupervised setup differs from the usual supervised learning setup, where cross-encoders almost always outperform bi-encoders.

-------------------------------
In the end, we summarise the changes we made to the paper:

1. We added *“Choice of loss functions: maintaining student-teacher discrepancy is the key.”*  in Section 2.3 as the intuition and justification of why choosing different loss functions for bi-to-cross and cross-to-bi distillations.
2. We added *“Gradient derivation of BCE and MSE loss.”* in Appendix A.2 to theoretically demonstrate why BCE is a more *forgiving* loss and less likely to cause overfitting of cross-encoders.
3. We added *“Loss function configurations.”* in Appendix A.2 and Table 8 to demonstrate our empirical findings of other choices of loss configurations.
4. We added *“Do sentence pairs leak label information?”* in Appendix A.2 and Table 9 to discuss why correlated sentence pairs provided by the task do not reveal any information to what is tested in the task. Table 9 lists the model performance of contrastive methods with task sentence pairs as positive pairs.
5. We added Appendix A.3 and Table 10 to thoroughly discuss training overhead and tested the model across different devices.
6. We extended Table 4 & 5 in appendix to include two more contrastive base models, i.e. DeCLUTR and Contrastive-Tension, to validate the robustness of the approach.

-------------------------------
Code to replicate Trans-Encoder can be found [here](https://drive.google.com/file/d/14Ufp7o_BvWBE1Ip-uuPWZIJHnaY-Bppa/view?usp=sharing).

---

### Decision · Program_Chairs · 2022-01-20

**Decision:**

Accept (Poster)

**Comment:**

For pairs of pieces of text, the central idea of this paper is to combine the approaches of using bi-encoders (where a vector is formed from each text then compared), which are easily trained in an unsupervised manner, with cross-encoders (where the two texts are related at the token level), which are normally trained in a supervised manner. The chief contribution of this paper is to train a combined model (as a "trans-encoder") by doing co-training of both model types in an alternating cyclic self-distillation framework. The paper suggests that this allows unsupervised training of a cross-encoder. This claim met some pushback from the reviewers, since the method does require good quality aligned text pairs (much like a traditional MT system does), and so the result is a  task-specific sentence-pair modeling approach rather than a generic unsupervised learning approach.

In the discussion, downsides included the claims of "unsupervised" being overstated, the genuine remaining need for related sentence pairs, the lack of a more theoretical understanding of why this works, and the feeling that the paper is not yet fully mature. Upsides include solid work building from existing models, big performance improvements over SimCSE, novelty in combining previous ideas in a new way for a new problem, and good experiments. To my mind, while the requirement of related sentence pairs does mean the model is task-specific and less than fully unsupervised, this is still a common and useful scenario, the performance of the model is strong, and, while the proposed model is built from existing components and ideas, they are combined in an interesting new way to achieve an intriguing and strong new way of training models, and the discussion here (and now in Appendix A.2) of what the authors had to do to get the model to work in terms of choosing different losses, etc., convincingly demonstrated that the authors had thought significantly and deeply about the nature of their proposal and how to get it to work well. Moreover the authors were able to work expeditiously during the reviewing period to address other weaknesses, such as now providing results with other methods than SimCSE (DeCLUTR and Contrastive Tension) and on other language models (RoBERTa).

As such, although this paper is clearly somewhat borderline rather than an unambiguous accept, I find myself quite convinced by the novelty, thoroughness, and intriguing nature of this work, and so my vote is to accept it.